# Degradation graphs reveal hidden proteolytic activity in peptidomes

Erik Hartman[1]*, Johan Malmström[1], Jonas Wallin[2]

**1** Division of Infection Medicine, Faculty of Medicine, Lund University, Lund, Sweden, **2** Department of Statistics, Lund University, Lund, Sweden

\* erik.hartman@med.lu.se

## Abstract

Protein degradation is a regulated process that reshapes the proteome and generates bioactive peptides. Peptidomics and degradomics enables large-scale measurement of these peptides, yet most data analyses approaches treat peptides as isolated endpoints rather than intermediates produced by sequential cleavage. Here, we introduce degradation graphs, a probabilistic framework that represents proteolysis as a directed acyclic network of cleavage events with explicit absorption. From single-snapshot peptidomes, we infer graph weights by gradient descent or linear-flow optimization, quantify flows through branches and bottlenecks, and correct a core bias in conventional quantification. Across three biological datasets, failure to model downstream trimming leads to 3–4-fold underestimation of upstream proteolytic activity. Moreover, degradation graphs provide graph-structured features that enable machine learning models to capture protease-specific signatures from both graph topology and sequence context. Taken together, these findings establish explicit degradation modeling as a practical approach to mechanistic and interpretable peptidomics, bridging the fields of degradomics and peptidomics.

## Author summary

Proteins are continuously broken down into smaller fragments called peptides, a process known as proteolysis. This controlled degradation shapes the proteome, regulates signaling, and generates bioactive molecules involved in immunity, inflammation, and disease. Modern mass spectrometry techniques can measure thousands of such peptides, yet most analytical methods treat these peptides as isolated snapshots rather than as part of an ongoing proteolytic process.

In this study, we introduce a framework for modelling protein degradation as a networked process. We use **degradation graphs** to represent how proteins break down step by step, where each peptide is connected to all fragments it can generate. From experimental peptidomic data, these graphs can be reconstructed to quantify how

**Data availability statement:** The datasets used in this study are available from ProteomeXchange with the following IDs: PXD037803, PXD012210 and PXD048892. The code repository for this project is available at GitHub.

**Funding:** JM was funded by The Swedish Research Council (F2024/2118), Xinnate AB (V2023/2285), Tanea AB (F2019/1184), and Novo Nordisk (F2025/123). The funders had no role in study design, data collection and analysis, decision to publish, or preparation of the manuscript.

**Competing interests:** The authors have declared that no competing interests exist.

proteolytic activity flows through the network, allowing more accurate estimation of enzyme activity and identification of bottlenecks in the degradation process. Using both experimental and clinical datasets, we show that traditional analyses underestimate total proteolytic activity by up to fourfold. By treating the peptidome as a dynamic system rather than a static collection of fragments, degradation graphs bridge the gap between peptidomics and degradomics, offering a mechanistic and interpretable view of how proteins are proteolytically processed in health and disease.

## 1 Introduction

Protein degradation is central to cellular life. Beyond recycling proteins to maintain homeostasis, proteolysis regulates protein localization, abundance, and activity, and acts as an irreversible post-translational modification [30,37] (Fig 1a). Proteolysis also generates peptides with functional roles, including hormones [7], neurotransmitters [41,48], antimicrobial peptides [15,36,50,56], and modulators of inflammation [6,17,50]. These bioactive fragments show that degradation is not a terminal process but a dynamic one that remodels the proteome while creating new biological activities.

The importance of proteolysis is especially evident in host–pathogen interactions, where both host and pathogen exploit proteases to gain an advantage [13]. Host proteases release antimicrobial peptides, modulate inflammation, and degrade virulence factors [2,8,35,45], while pathogens secrete proteases that disable antibodies, cytokines, and complement proteins [8,9,12,19,20,23,24,35,40,42,43,46,47,49,50, 53] (Fig 1b). These examples illustrate that proteolysis is tightly regulated, context dependent, and mechanistically complex.

Two fields attempt to disentangle this complexity. Degradomics focuses on protease activities, cleavage sites, and substrate profiles, with applications to cancer [26,32,34,52], neurodegeneration [1,27], and inflammation [4,5,28,31,32,37]. Peptidomics emphasizes large-scale identification and quantification of endogenous peptides for biomarker discovery and activity profiling [10,16,18,25,29,38]. Both rely on mass spectrometry, but degradomics enriches for neo-termini [21,44,54], whereas peptidomics quantifies peptides globally [10] (Fig 1c, 1d). Despite methodological differences, both aim to understand how proteolysis shapes the peptidome.

Current data analysis strategies in degradomics and peptidomics treat peptides as static endpoints, although peptides are intermediates that undergo further proteolysis. Only a few studies have attempted to model this process. Yi et al. showed that exoprotease activity can be represented as a sequential multistep reaction using isotopically labelled fibrinogen peptide A and time-resolved quantification of its degradation products [55]. Kluge et al. introduced the first formal framework for modelling exoprotease activity as a Markov process on a graph in which edges represent cleavage events and parameters correspond to amino acid specific degradation rates [22]. Aiche et al. extended this framework to endoproteolytic activity and estimated kinetic parameters from time series data [3]. These studies provide a conceptual basis and

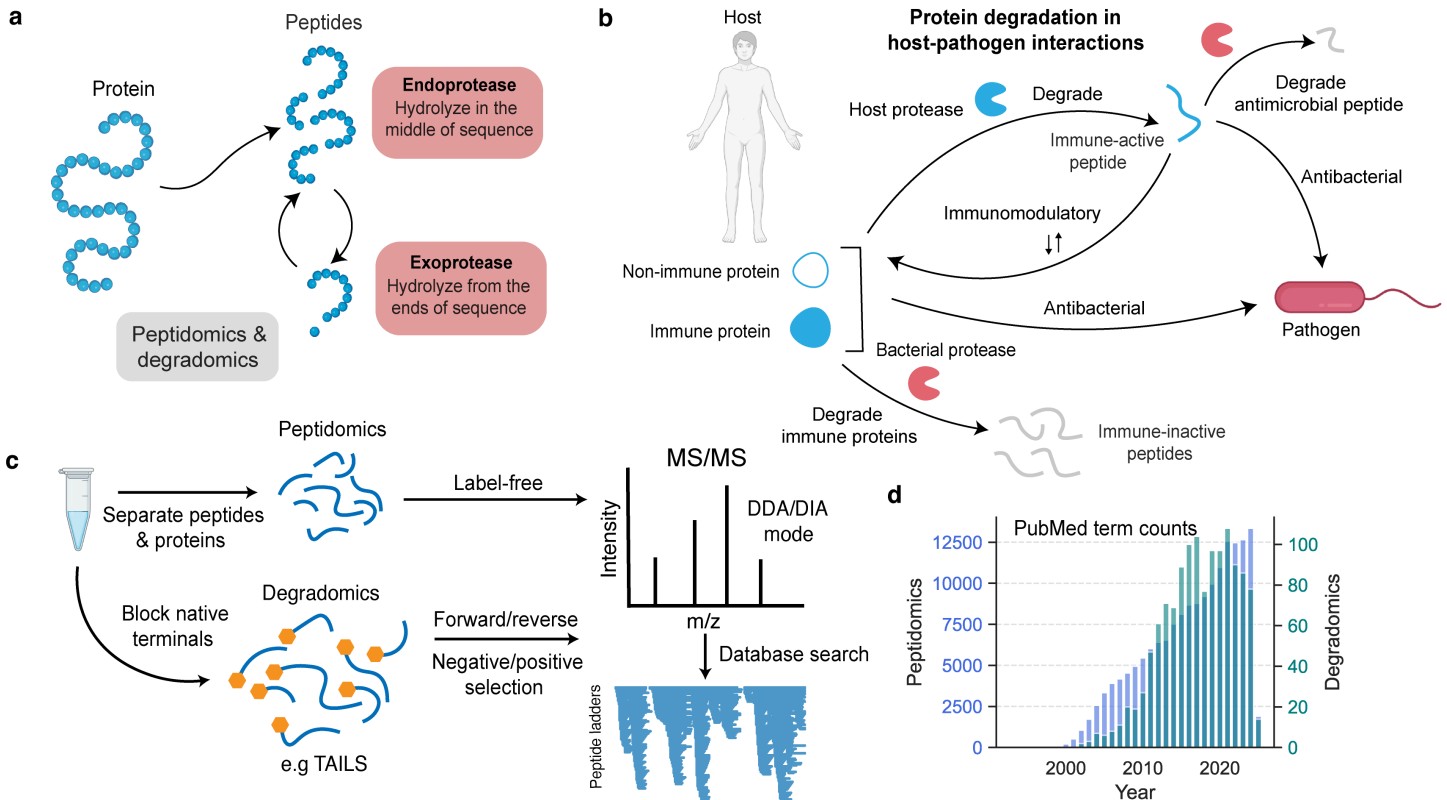

**Fig 1**. **Overview of proteolysis and peptide-based degradomic analysis. a** Proteins are hydrolyzed into peptides by proteases with distinct cleavage modes. Endoproteases cleave internal peptide bonds, whereas exoproteases trim residues sequentially from peptide termini. Both contribute to continuous peptide turnover. **b** During host–pathogen interactions, proteolysis shapes immune responses on both sides. Host proteases can release antimicrobial and immunomodulatory peptides, while bacterial proteases degrade host defense and immune proteins, generating immune-inactive fragments. The balance of these processes determines the outcome of infection. **c** Peptidomics and degradomics use mass spectrometry to characterize proteolys......is from complementary angles. Peptidomics quantifies endogenous peptides directly, whereas degradomics, using methods such as TAILS, blocks native termini to enrich for neo-termini generated by protease activity. Peptides are analyzed by LC–MS/MS in DDA or DIA mode, searched against protein databases, and visualized as peptide ladders. **d** Growth of peptidomics and degradomics research over time, shown as PubMed term counts by year, illustrating the expansion of both fields, especially peptidomics. This figure was made with BioRender.

proof-of-concept for modelling protein degradation but remain restricted to exoproteolysis or depend on temporally resolved measurements that are rarely available. They also precede modern informatics workflows, which limits their applicability to large datasets. As a result, upstream protease activities are systematically underestimated and degradomics and peptidomics remain analytically disconnected.

Here, we introduce an alternative framing of **degradation graphs**: a representation of proteolysis as a directed acyclic graph in which nodes are peptides, edges represent cleavage events, and each node is assigned an explicit absorption probability. This framework treats peptides as dynamic intermediates and captures the sequential nature of degradation without the need for time-series data, making it a natural extension to prior works. By first formalizing degradation graphs probabilistically, we show how degradation graphs generate marginal peptide distributions, and establish an intuitive flow-based interpretation. Second, we present two methods to reconstruct degradation graphs from single-snapshot peptidomes: (i) gradient-based optimization with closed-form gradients, and (ii) a linear-flow formulation solved with linear programming. Third, we demonstrate applications of degradation graphs using simulated proteolysis and experimental

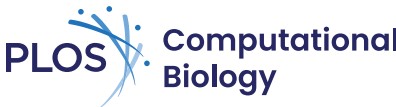

data, showing that degradation graphs unify peptidomic and degradomic goals by connecting observed peptides to the mechanistic processes that generate them.

## 2 Results

We introduce a framework that connects peptidomic observations to the degradomic processes that generate them. Rather than treating peptides as isolated endpoints, **degradation graphs** model their sequential connectivity through directed cleavage relationships. This representation captures both degradation routes and resulting peptide abundances, enabling mechanistic quantification, peptide clustering, and graph-based inference. In contrast to previous approaches, this method does not require time-resolved measurements and can model protein degradation by applying modern optimization techniques to large peptidomes generated with contemporary mass spectrometry workflows. Below, we formalize degradation graphs, show how they can be inferred from single-snapshot peptidomes, and demonstrate their applications to *in vitro* and *in vivo* data.

### 2.1 Formal definition of the degradation graph

A degradation graph is defined as a directed acyclic graph $G = (V, E)$, where each node $v \in V$ represents a peptide (including the full-length protein), and each directed edge, $(v \to u) \in E$, indicates that peptide $v$ can degrade into peptide $u$ (Fig 2a). This graph encodes the network of cleavages that progressively fragment the intact protein into smaller peptides. To quantify this process, each edge is assigned a transition probability $w_{v \to u}$ specifying the likelihood that $v$ degrades into $u$. The difference between one and the sum of outgoing probabilities represents the chance that $(v)$ remains stable, an

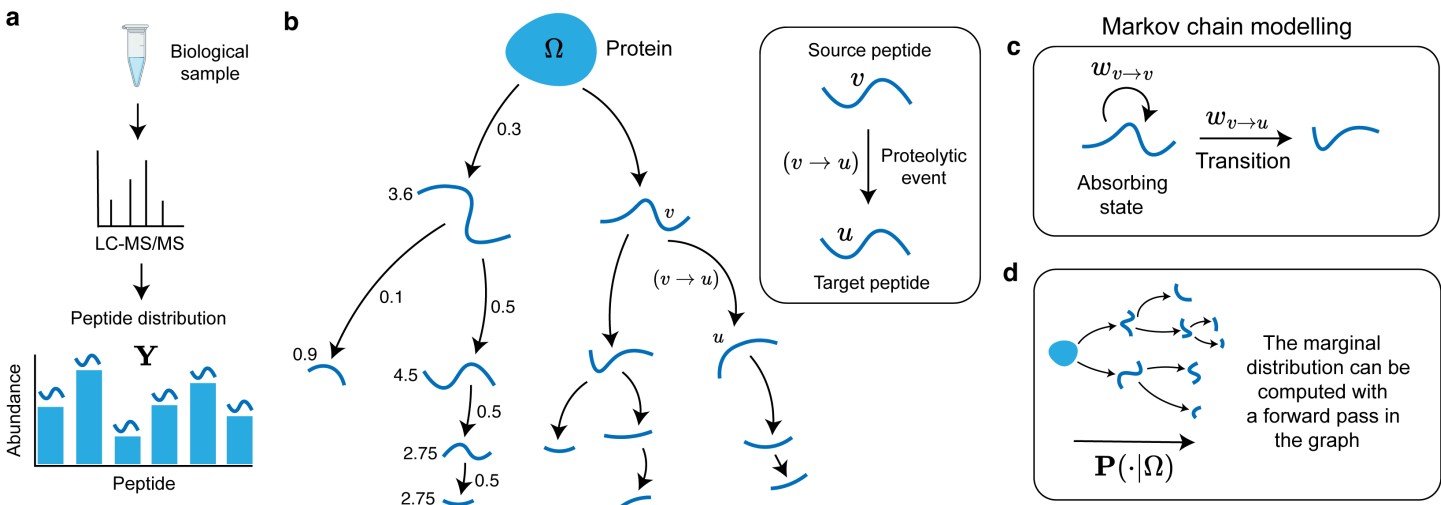

**Fig 2**. **Definition and probabilistic formulation of the degradation graph. a** A biological sample is analyzed with the mass spectrometer followed by peptide identification and quantification, resulting in a peptide distribution. **b** A degradation graph represents proteolysis as a directed acyclic graph in which each node corresponds to a peptide (including the intact protein, $\Omega$), and each directed edge $(v \to u)$ denotes a proteolytic event where peptide $v$ is cleaved into a shorter peptide $u$. The graph thereby encodes sequential cleavage relationships that describe how a protein is progressively degraded into smaller fragments. **b** Each edge is associated with a transition probability $w_{v \to u}$ describing the likelihood that $v$ degrades into $u$. The probability that $v$ remains intact, its absorption probability, is modeled as $w_{v \to v} = 1 - \sum_{u \in \mathcal{C}(v)} w_{v \to u}$, where $\mathcal{C}(v)$ is the set of child nodes. This defines a Markov chain in which transitions correspond to cleavage events and self-loops to peptide stability. **c** The overall peptide distribution, $\mathbf{P}(\cdot|\Omega)$, can be obtained by propagating probability mass from the protein node $\Omega$ through the graph in a forward pass. At each node, a fraction of the mass is absorbed, and the remainder distributed according to the outgoing transition probabilities, yielding a marginal distribution that reflects the steady-state abundances of peptides generated by the degradation process.

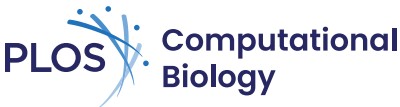

absorption event in the Markov sense, captured by a self-loop, $(v \to v)$, with a probability of

$$w_{v \to v} = 1 - \sum_{u \in \mathcal{C}(v)} w_{v \to u},$$

where $\mathcal{C}(v)$ denotes the set of child nodes. This probabilistic formulation extends earlier models of exoprotease activity [22] to general peptide degradation.

An intuitive way to interpret these probabilities is through the notion of flow. Beginning with probability mass of one at a protein node $\Omega$, mass flows through the graph according to the assigned edge probabilities (Fig 2b). As each node $v$ is reached, a proportion $w_{v \to v}$ of the mass arriving at $v$ is absorbed in $v$, and the remainder is distributed among its children $\{u \in \mathcal{C}(v)\}$ in proportion to $\{w_{v \to u}\}$. Because the graph contains no cycles except for self-loops, all mass is eventually absorbed in the nodes, generating a peptide distribution **P** (Fig 2c). The total absorbed mass for each node can be interpreted as its overall probability of being the endpoint of a degradation pathway starting at $\Omega$.

## 2.2 Identifying the degradation graph

Without high resolution time-series data, proteolysis cannot be observed directly and the degradation graph must be inferred from the measured peptide distribution (Fig 4a). Previous work reconstructed proteolysis kinetics from time-resolved data [3], but such data are rarely available from samples from *in vivo* experiments and from patients which constitute a large majority of peptidomics experiments. We therefore treat the graph as a latent structure inferred from a static peptidome snapshot, following the principle that the optimal graph should reproduce the empirical peptide distribution **Y**. Here, **Y** denotes the empirical peptide abundance distribution, with component **Y**($u$) corresponding to the observed abundance of peptide $u$.

We frame inference as an optimization problem. The model predicts a marginal peptide distribution **P** determined by the edge transition probabilities $w_{u \to v}$. These probabilities are adjusted to minimize a loss function $\mathcal{L}(\mathbf{Y}, \mathbf{P})$ measuring the discrepancy between modeled and observed abundances (Fig 2b). A softmax mapping enforces valid transition probabilities that sum to at most one per node, with the residual interpreted as absorption. Gradients $\partial \mathcal{L}/\partial w_{u \to v}$ are computed through **P**, and iterative updates refine the graph until its predicted distribution aligns with the data (Algorithm 1, Fig 3b–3c).

An equivalent and more intuitive formulation expresses the same problem as a flow system, in which each edge carries a non-negative flow $F_{u \to v}$ representing the probability mass transferred from $u$ to $v$. Mass conservation requires that the outgoing flow and absorbed mass at each node equal its total incoming mass. When these conditions are written as linear constraints, the result is a standard optimization problem that can be solved with linear programming. The solution, provides flows $\{F_{u \to v}\}$ whose ratios directly yield transition probabilities $w_{u \to v} = F_{u \to v}/(\sum_x F_{x \to u})$ (Algorithm 2, Fig 4b).

The identification of a degradation graph is an underdetermined problem because different graph structures can produce the same observed peptide distribution (see **Discussion** and **Methods: Formalization of key limitations and assumptions**). To assess how much the inferred graphs vary, we ran gradient descent from different random initializations (five replicates) across graphs of increasing size. We measured variation in edge weights using their coefficient of variation. Although the mean variation increased with graph size, it remained below 10 percent even for the largest graphs (Fig 3d).

To illustrate the inference procedure on biological datasets, we reconstructed the degradation graph of trypsinized $\beta$-actin from human cell lysates (Fig 3e–3g). Peptide intensities were averaged across samples, and edge weights were optimized by gradient descent. The inferred graph recapitulated the expected degradation hierarchy, capturing sequential fragmentation from the intact protein toward shorter peptides. This demonstrates that degradation graphs can be recovered from single-snapshot peptidomes, providing a compact representation of degradation connectivity.

**Algorithm 1.  Gradient-based inference of transition probabilities $w_{u \to v}$.**

**Input:** `Degradation graph` $G = (V, E)$`; root (protein)` $\Omega$`; observed peptide distribution` **Y**`; learning rate` $\eta$`; epochs` $T$

**Output:** `Transition probabilities` $w_{u \to v}$ `for all` $(u, v) \in E$ `and absorptions` $w_{u \to u}$`; predicted distribution` **P**

1 **Parameterization:** `Each node` $u$ `has trainable logits` $\theta_{u \to v}$ `for` $v \in \mathcal{C}(u) \cup \{u\}$`. Transition probabilities are given by a softmax normalization:`

$$w_{u \to v} = \frac{e^{\theta_{u \to v}}}{\sum_{x \in \mathcal{C}(u) \cup \{u\}} e^{\theta_{u \to x}}}, \qquad \sum_{v \in \mathcal{C}(u) \cup \{u\}} w_{u \to v} = 1.$$

2 **for** $t = 1$ **to** $T$ **do**

 `// Forward pass (propagate mass through the DAG)`

3 `Initialize` $p(u) \leftarrow 0$ `for all` $u \in V$; $p(\Omega) \leftarrow 1$.

4 `Initialize` **P**$(u) \leftarrow 0$ `for all` $u \in V$.

5 **for** *u in topological order of G* **do**

6 **P**$(u) \leftarrow$ **P**$(u) + p(u)\, w_{u \to u}$ `// absorbed mass`

7 **for** $v \in \mathcal{C}(u)$ **do**

8 $p(v) \leftarrow p(v) + p(u)\, w_{u \to v}$ `// propagate remaining mass`

 `// Loss and gradient update`

9 $\mathcal{L}(\mathbf{Y}, \mathbf{P}) \leftarrow \sum_{u \in V} (\mathbf{P}(u) - \mathbf{Y}(u))^2$

10 $\theta_{u \to v} \leftarrow \theta_{u \to v} - \eta\, \nabla_{\theta_{u \to v}} \mathcal{L}$ `for all` $(u, v) \in E$

Both optimization schemes are implemented in modern frameworks like PyTorch [39] and PuLP [33] are available in our GitHub repository. The advantage of the LP formulation is fast and guarantees exact satisfaction of the flow constraints. While gradient descent is slower, it is more flexible because the objective is written as a differentiable loss, which allows future extensions such as regularization terms, priors, or custom penalties that guide the solution toward preferred graph structures. This flexibility makes gradient descent well suited for exploring alternative modeling assumptions that cannot be expressed cleanly using linear constraints. In practice, the runtime remains short even for large peptidomes, so the additional compute required for gradient descent is minimal. For this reason, we view gradient descent as the default choice (Table 1).

## 2.3 Why degradation graphs matter

Traditional peptidomic quantification assumes that each peptide reflects a single proteolytic event, and the proteolytic activity is therefore simply calculated as the sum of peptide abundances: $\sum_{v \in V \setminus \Omega} Y(v)$. In reality, peptides can themselves act as substrates for further degradation, as modelled by Yi et al. [55], meaning that their observed abundance underestimates the upstream activity that generated them. In conventional workflows, only the direct link between the protein, $\Omega$, and its peptide, $v$, is considered, while degradation of $v$ into $u$ is ignored (Fig 4a). Consequently, the apparent cleavage weight $w_1$ derived from peptide abundance alone is smaller than the true value $w_2$ obtained when downstream trimming is modeled. The total proteolytic events in a degradation graph can therefore be computed as the sum of flows, and the ratio by which classical modelling underestimates the proteolytic activity, $\Delta$, can be computed as:

$$\Delta = \frac{\sum_{(u \to v) \in E} F_{u \to v}}{\sum_{v \in V \setminus \Omega} Y(v)},$$

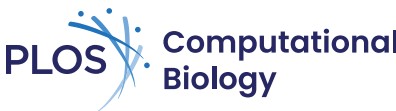

**Fig 3**. **Inferring degradation graphs from observed peptidomes. a** The degradation graph is a latent structure that describes the proteolytic relationships giving rise to the observed peptidome. Observed peptide abundances are used to infer the most plausible graph structure and transition probabilities. **b** Inference is formulated as an optimization problem in which the modeled marginal peptide distribution is fitted to the measured distribution. Edge transition probabilities are iteratively updated by gradient descent to minimize the loss $\mathcal{L}(\mathbf{Y}, \mathbf{P})$, or equivalently inferred as a linear-flow system solved by linear programming under mass-conservation constraints. **c** Example of graph optimization by gradient descent. The mean-squared-error loss (blue) decreases over iterations as the inferred graph converges toward the true degradation graph, measured by the total deviation of edge

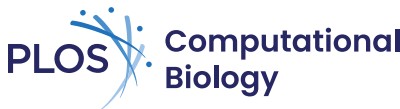

weights (orange). **d** The coefficient of variation of edge weights when applying gradient descent to graphs of increasing sizes. The shaded region shows ± 1 standard deviation. **e** Experimental validation using *in vitro* trypsin digestion of human pharyngeal epithelial cells (Detroit 562; dataset PXD037803 [14]). Peptides were identified and quantified by LC–MS/MS and analyzed to reconstruct the underlying degradation graph. **f** Peptides mapped onto the $\beta$-actin backbone and colored by their number of descendants in the inferred graph, illustrating hierarchical fragmentation patterns. **g** Relationship between peptide abundance and total inflow ($\sum$inflow) for each peptide. The solid line indicates where the modeled inflow and measured abundance are equivalent, demonstrating quantitative agreement between the inferred degradation flow and experimental intensities.

**Algorithm 2. Linear-flow reconstruction of degradation graphs.**

**Input:** Degradation graph $G = (V, E)$; root (protein) $\Omega$; observed peptide distribution $\mathbf{Y}$
**Output:** Feasible edge flows $F_{u \to v} \geq 0$ and transition probabilities $w_{u \to v}$

1 **Variables:** Define a non-negative flow variable $F_{u \to v} \geq 0$ for each $(u, v) \in E$.
2 **Root injection:**

$$\sum_{v \in \mathcal{C}(\Omega)} F_{\Omega \to v} = 1.$$

3 **Mass conservation:** For every node $u \in V \setminus \{\Omega\}$,

$$\sum_{v \in \mathcal{C}(u)} F_{u \to v} + \mathbf{Y}(u) = \sum_{x \in \mathcal{P}(u)} F_{x \to u}.$$

4 **Solve LP:** Find a feasible $\{F_{u \to v}\}$ satisfying the constraints above (or minimize a small regularizer, e.g. $\min \sum_{(u,v) \in E} F_{u \to v}$).
5 **Convert to transition probabilities:** For each node $u$, define its total incoming mass $M_u = \sum_{x \in \mathcal{P}(u)} F_{x \to u}$, and set

$$w_{u \to v} = \begin{cases} \dfrac{F_{u \to v}}{M_u}, & M_u > 0, \\ 0, & M_u = 0, \end{cases} \quad \forall v \in \mathcal{C}(u), \qquad w_{u \to u} = 1 - \sum_{v \in \mathcal{C}(u)} w_{u \to v}.$$

In the $\beta$-actin data, this led to a roughly two-fold underestimation of total proteolytic activity (Fig 3f). We examined how the underestimation ratio changes as a function of degradation extent in a proteolysis simulation. As simulation time increased and more cleavages accumulated, the ratio rose (S1 Fig). This behavior is expected, as larger degradation graphs contain longer and more branched pathways, and simply summing observed peptide intensities fails to capture the cumulative loss of abundance along these routes. These results show quantitatively that conventional peptide-summation methods increasingly underestimate proteolytic activity as the peptidome becomes more complex.

By explicitly modeling the sequential flow of degradation, degradation graphs correct this bias. Each peptide's inflow equals the mass absorbed in that peptide plus the flow routed to its descendants, ensuring that upstream events are represented in proportion to their true activity. Beyond improving quantitative accuracy, the graph structure introduces interpretable organization into the peptidome (Fig 4b–4c). Related peptides naturally cluster into branches that reflect shared ancestry, while intermediate nodes identify points of control where flow divides or accumulates. These network features provide biologically meaningful units, branches and bottlenecks, that are more stable and interpretable than single peptide intensities and can be linked directly to specific protease families or pathways.

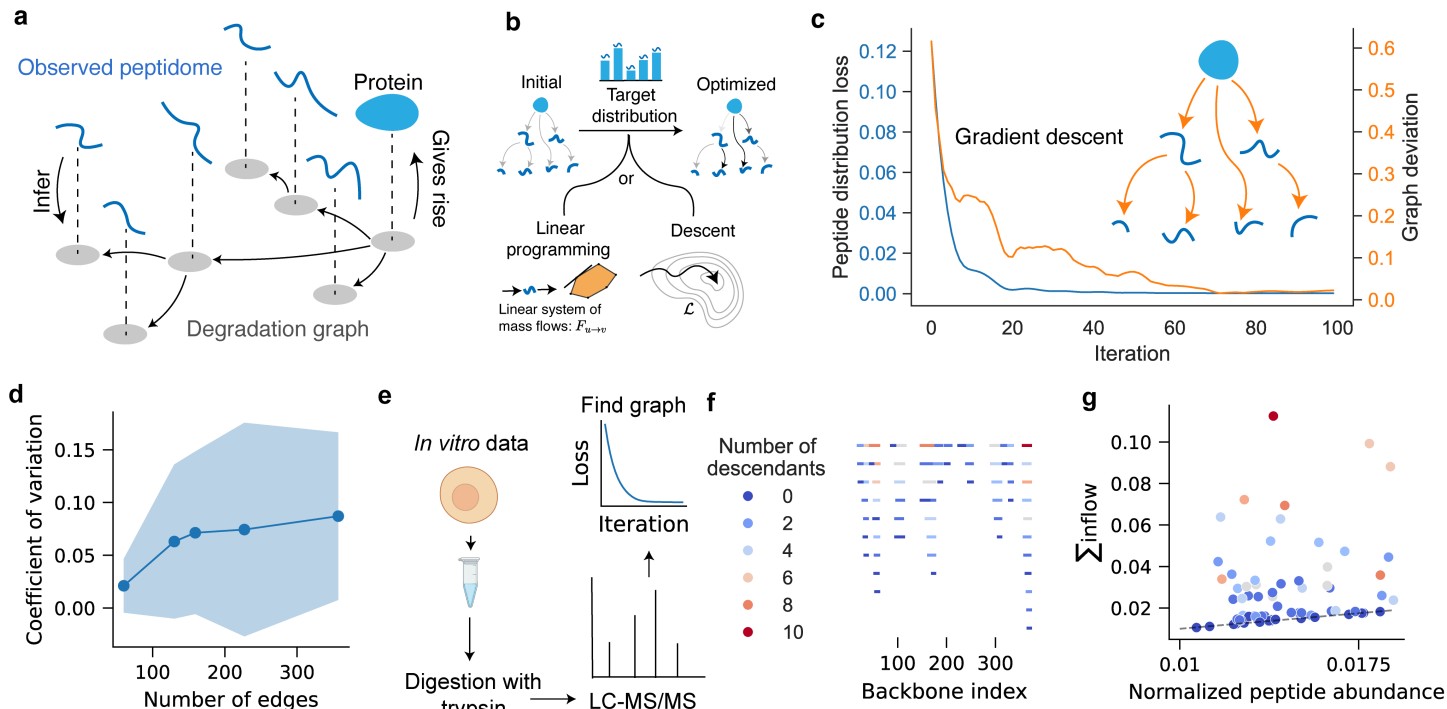

**Fig 4**. **Why degradation graphs improve protease quantification and enable predictive modeling. a** Conventional peptidomic analyses assume that each peptide arises directly from the parent protein, neglecting that peptides can themselves undergo further degradation. This omission leads to systematic underestimation of upstream proteolytic activity. In degradation graphs, downstream cleavages (e.g., $v \to u$) are explicitly modeled, revealing the true effective transition weight ($w_2 > w_1$) and correcting activity quantification. **b** By representing peptide abundances as probabilistic flow through the graph, degradation graphs enable *flow quantification*, where the total upstream activity can be mapped to specific proteases using databases such as MEROPS or TopFIND. **c** *Branch quantification* summarizes local subgraphs, sets of related peptides sharing a degradation ancestry, into stable and interpretable biological units. Bottlenecks and branch points identify sites of proteolytic control or preferential cleavage. **d** The explicit graph topology also permits machine learning applications. Degradation graphs generated *in silico* for four proteins (beta actin, hemoglobin subunit beta, thrombin and apolipoprotein a1) digested by trypsin or elastase were encoded as graph structures and classified using a GraphConv neural network trained on node (position, abundance, length) and edge (transition weight) features. **e** Left: receiver operating characteristic (ROC) curves showing high classification performance across proteins (overall ROC–AUC = 0.915). Right: Kernel density estimates of predicted probabilities for trypsin- and elastase-derived graphs in the validation set, illustrating accurate separation of protease-specific degradation patterns. **f** Peptidome plots of $\beta$-actin in cell lysates digested with different enzymes. **g** ROC of the GraphConv network trained to identify the enzyme.

**Table 1**. **Runtime of gradient descent and linear programming solvers across increasing graph sizes.**

| # edges | gradient descent (s, mean $\pm$ std) | LP (s, mean $\pm$ std) |
|---|---|---|
| 24 | 1.60 $\pm$ 0.26 | 0.031 $\pm$ 0.017 |
| 46 | 2.59 $\pm$ 0.02 | 0.0229 $\pm$ 0.0017 |
| 68 | 3.79 $\pm$ 0.04 | 0.0235 $\pm$ 0.0021 |
| 128 | 6.22 $\pm$ 0.01 | 0.0261 $\pm$ 0.0037 |
| 253 | 10.97 $\pm$ 0.14 | 0.0376 $\pm$ 0.0137 |
| 360 | 15.95 $\pm$ 0.24 | 0.0540 $\pm$ 0.0150 |

Finally, the explicit graph topology enables graph-based inference (Fig 4d). Node and edge-level features such as position, length, abundance, and transition probability can be integrated into machine learning models that recognize characteristic degradation regimes. As an example, degradation graphs derived from *in silico* proteolysis of several proteins

were classified by a GraphConv network, which separated trypsin from elastase-derived graphs with high accuracy (over-all ROC–AUC = 0.915, Fig 4e). We also applied the GraphConv network on cell lysate peptidomes when digested with different enzymes. $\beta$-actin was chosen due to its high coverage in the samples, and the graphs were identified through gradient descent (Fig 4f). On these samples, the model achieves a near perfect ROC-AUC (Fig 4g). While these demon-strations are limited to simulated data and simple digestions, it demonstrates the concept that degradation graphs not only correct quantification but also provide a foundation for predictive analysis of protease activity.

### 2.4 Applications to *in vivo* data

To investigate the performance of degradation graphs on clinical data, we analyzed a dataset of urinary peptides from diabetic and healthy individuals [51] (Fig 5a). Uromodulin, a previously proposed biomarker candidate for diabetes, was selected as a case study. For each sample, degradation graphs were reconstructed by gradient descent to align modeled and observed peptide abundances (Fig 5b). Without accounting for downstream degradation, conventional quantification underestimated total proteolytic activity by about 3.5-fold (CI 95%: 3.52-3.58, Fig 5c). No significant difference in under-estimation ratios was found between healthy and diabetic patients. Mapping total inflow along the uromodulin backbone revealed differential degradation concentrated in the C-terminal biomarker region (Fig 5d).

In a second campaign we analyzed peptidomic data from porcine wound fluids infected with either *Staphylococcus aureus* or *Pseudomonas aeruginosa* [11] (Fig 6a). Previous work identified the N-terminal region of hemoglobin subunit

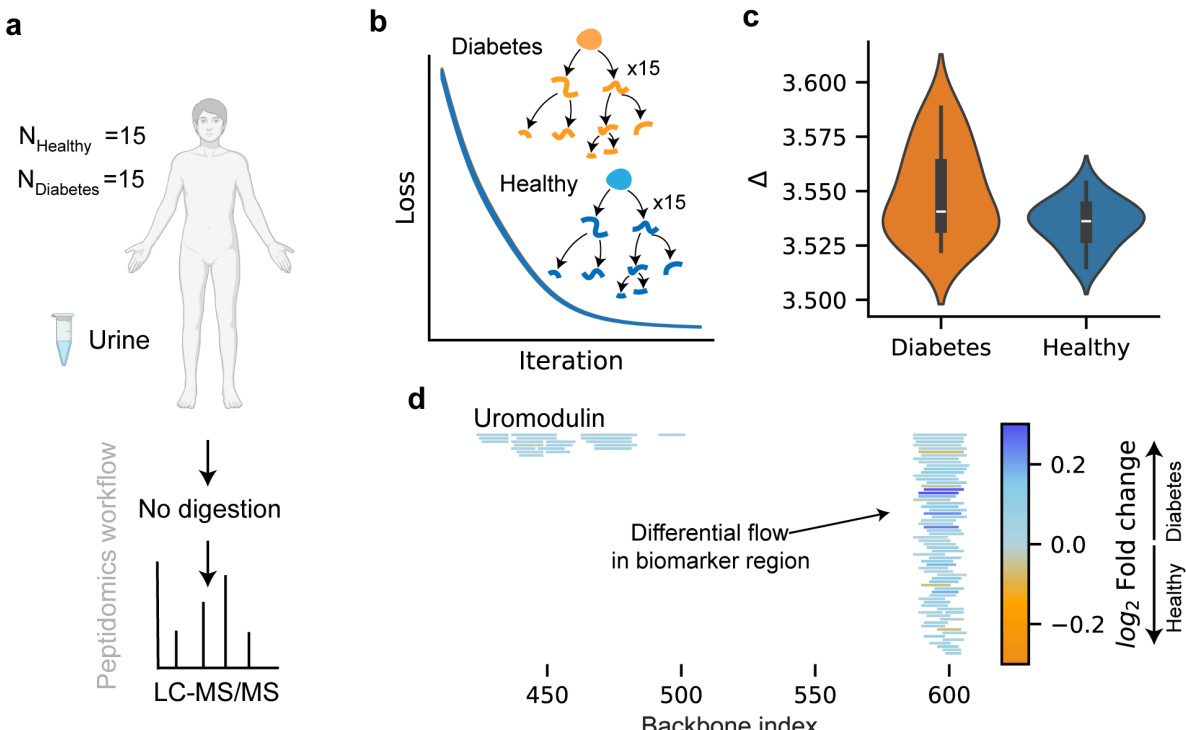

**Fig 5**. **Degradation graph analysis of urinary peptidomes from diabetic and healthy individuals. a** Peptidomic data from urine samples of indi-viduals with diabetes ($n = 15$) and healthy controls ($n = 15$) [51] were analyzed without enzymatic digestion using LC–MS/MS. **b** For each sample, degradation graphs were reconstructed by gradient descent to fit the modeled peptide distribution to observed abundances. **c** Comparison of total gen-erated versus observed peptide abundance revealed that conventional quantification underestimated proteolytic activity by approximately 3.7-fold in both groups. **d** Mapping total inflow along the uromodulin sequence showed a localized increase in degradation flow within the biomarker-associated region, highlighting differential proteolysis between healthy and diabetic samples. This figure was made with BioRender.

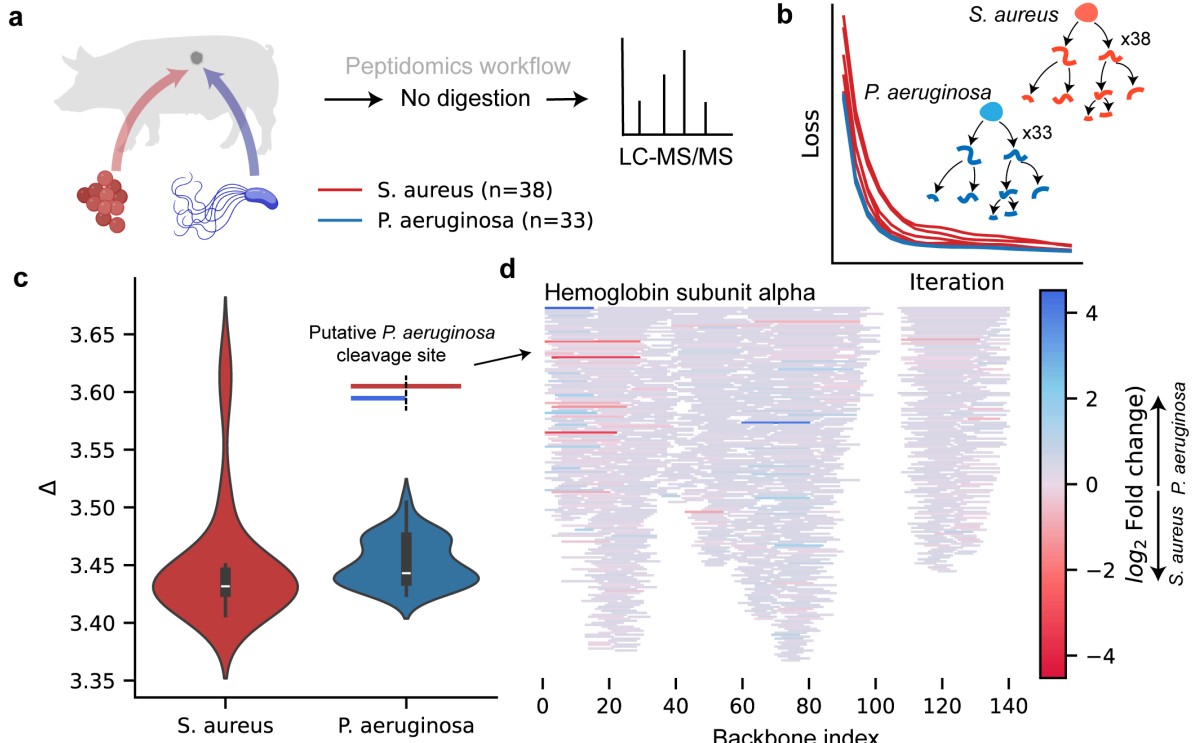

**Fig 6**. **Degradation graph analysis of porcine wound fluid peptidomes from bacterial infections. a** Peptidomic data from porcine wound fluids infected with *Staphylococcus aureus* (*n* = 38) or *Pseudomonas aeruginosa* (*n* = 33) [11] were analyzed without enzymatic digestion using LC–MS/MS. **b** For each sample, edge transition probabilities were optimized by gradient descent to reproduce measured peptide distributions. **c** The ratio between generated and observed peptide abundance showed that neglecting sequential degradation led to an average 3.5-fold underestimation of total proteolytic activity. **d** Visualization of total inflow along the hemoglobin subunit alpha backbone revealed differential degradation flow between infection types, with pronounced variation in the N-terminal biomarker region and an additional differential site around residues 60–80. This figure was made with BioRender.

alpha as a discriminatory biomarker between infection caused by the two different pathogens [16]. Degradation graph analysis reproduced this signal, identifying the key cleavage point in wounds infected by *P. aeruginosa*, and further high-lighted additional differential flow around residues 60–80 (Fig 6d). The total proteolytic activity was again underestimated by approximately 3.5-fold (CI 95%: 3.41-3.60) when sequential degradation was ignored (Fig 6c). No significant difference in underestimation ratios was found between *S. aureus* and *P. aeruginosa* infected wounds. These showcases highlight how degradation graph modelling can be used to improve peptidomic analyses.

We tested how sensitive the method is to training settings by sweeping over learning rates and numbers of epochs when applying gradient descent on a random sample from the clinical datasets. For both datasets and peptidomes, when the loss converged, the underestimation ratio ($\Delta$) increased and stabilized. The peptides with the highest inflow and the edges with the highest flows were recovered consistently across runs, especially in the smaller UMOD graph. The larger HBA graph showed more variation but still produced stable top features once training had converged (Fig 7a, 7b).

## 3 Discussion

This work reframes peptidomic analysis by treating peptides as intermediates in degradation pathways instead of termi-nal observations. Conventional proteolysis quantification collapses sequential proteolysis into static peptide abundances, leading to systematic underestimation of upstream enzymatic activity. By explicit modelling of degradation graphs, we

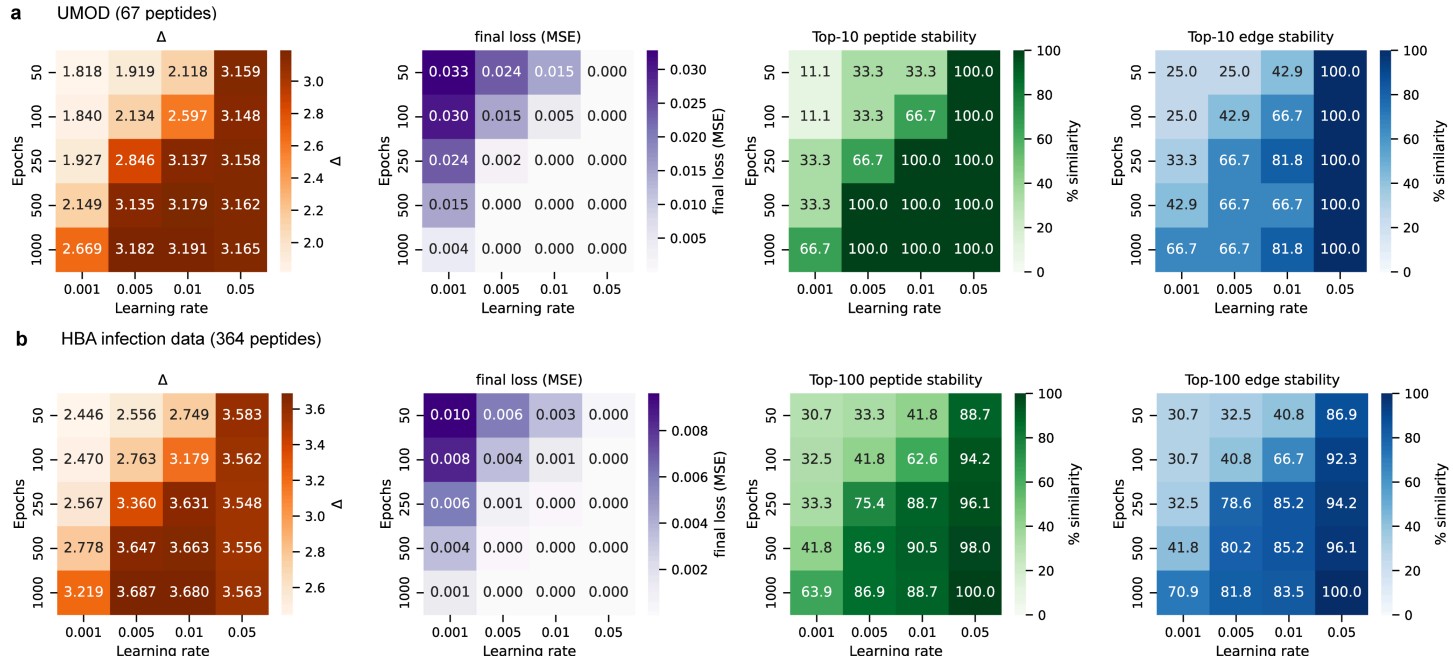

**Fig 7**. **Robustness of graph identification.** Parameter sweeps across learning rates and training epochs were used to assess the stability of degradation graph inference for the UMOD dataset (**a**) and the HBA infection dataset (**b**). Heatmaps show the underestimation ratio (Δ), final mean squared error, and the stability of the top peptides and edges when ranked of inflow and flow respectively, measured as the percentage of overlap across replicate runs.

recover the mechanistic flow of proteolysis. This structure brings several advantages such as improved quantification, where accounting for downstream degradation increased inferred proteolytic activity by roughly 3 to 4-fold. Beyond correcting quantification, degradation graphs provide a natural organizational layer for the peptidome. Branches summarize coherent degradation routes, while bottlenecks highlight rate-limiting intermediates or potentially bioactive fragments. The flow-based representation links degradomic logic to peptidomic observables, thereby unifying the two fields in a single probabilistic formalism.

Because degradation graphs encode proteolysis as a directed acyclic graph with explicit node and edge features, they can be used directly in graph neural networks and other machine learning models that are designed to learn from graph-structured data, unlike earlier approaches based solely on peptide intensities. In our demonstration, a simple Graph-Conv architecture could classify degradation graphs with topological and sequence-derived features alone. Although we only present this application as a proof-of-concept, it opens the door to predictive models that learn protease signatures, degradation kinetics, or tissue-specific proteolytic patterns directly from data.

In practical terms, degradation graphs enable richer analysis of static peptidomes without requiring time-resolved measurements. The approach can be scaled to proteome-wide reconstructions and integrated with protease databases such as MEROPS to identify enzyme families contributing to observed degradation flows. Applications include biomarker discovery, assessment of proteolytic state, and mechanistic stratification of diseases characterized by dysregulated proteolysis.

### 3.1 Limitations and outlook

There are two major challenges in building degradation graphs from peptidomic data. The first is that the graphs are not unique as different graph structures can produce the same marginal distribution, so the problem is inherently

underdetermined. Even so, our results show that the graphs inferred are reproducible across runs, with consistent identification of edges with high flow and nodes with high cumulative inflow.

The second challenge is inherited from the field of mass spectrometry peptidomics at large, which is the issue with peptide identification and quantification. Mass spectrometry fails to identify and quantify peptides for many reasons, including ionization efficiency, length, charge, or low abundance (see **Methods: Formalization of key limitations and assumptions**). If key intermediates are not detected, we do not have the true peptide distribution, and no method can recover a completely correct graph from incomplete data. Missing peptides can hide real branches in the degradation pathway or create apparent shortcuts that do not occur in biology. However, there are practical developments that could utilize the graph structure to help mitigate this issue. For example, imputation methods designed around degradation logic could infer likely missing intermediates, for example by enforcing that a parent fragment should give rise to two child fragments with compatible abundances. Further, sensitivity analyses where we add plausible missing peptides and examine how the graph changes, could show how strongly the results depend on detection gaps.

The severity of the detection problem will lessen as mass spectrometry instrumentation and workflows continue to improve, expanding the portion of the peptidome that can be reliably measured. The non-uniqueness issue will benefit from growing databases of known protease behavior and cleavage patterns. Together, these developments will make degradation graphs more accurate and more useful for understanding proteolysis in complex samples.

In summary, degradation graphs represent a step toward mechanistic, interpretable peptidomics. By embedding proteolysis into a quantitative, graph-based formalism, they connect sequence-level events to system-level outcomes. In doing so, it becomes possible to correct existing quantification biases providing conceptual bridge between degradomics and peptidomics, linking the enzymes that act and the fragments they leave behind.

## 4 Methods

### 4.1 Variable definitions

See Table 2.

### 4.2 Identifying the degradation graph using gradient-based optimization

We infer transition probabilities $w_{u \to v}$ by optimizing the degradation graph so that the modeled peptide distribution **P** reproduces the observed distribution **Y**.

Each node $u$ is parameterized by a vector of logits $\theta_u$, corresponding to transitions toward its child nodes $\mathcal{C}(u)$ and its self-absorption. A softmax transformation ensures that outgoing transition probabilities are valid and sum to one:

$$w_{u \to v} = \frac{\exp(\theta_{u \to v})}{\exp(\theta_{u \to u}) + \sum_{x \in \mathcal{C}(u)} \exp(\theta_{u \to x})},$$

where $\theta_{u \to u}$ represents the absorption parameter for node $u$.

Given an initial mass of 1 at the root $\Omega$, probability mass propagates through the graph in topological order. At each node $u$, the incoming probability mass $p(u)$ is distributed to its children and absorbed according to $w_{u \to v}$:

$$p(v) = p(v) + p(u) \, w_{u \to v}, \quad \forall v \in \mathcal{C}(u),$$

and the absorbed probability at node $u$ is

$$P(u) = p(u) \, w_{u \to u}.$$

Collectively, these absorption probabilities define the predicted peptide distribution **P**.

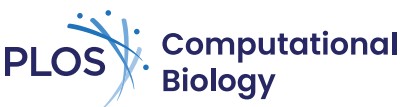

**Table 2**. Variable definitions.

| Variable | Definition |
|---|---|
| $G$ | Degradation graph. |
| $V$ | Set of vertices (nodes) in the degradation graph. Each node corresponds to a peptide sequence. |
| $E$ | Set of directed edges in the degradation graph, where each edge $(v \to u)$ represents a possible degradation event. |
| $w_{v \to u}$ | Transition probability from peptide $v$ to peptide $u$. Represents the probability mass transferred in one degradation step. |
| $F_{v \to u}$ | Flow through edge $(v \to u)$, representing the total probability mass passing from $v$ to $u$. |
| $\Omega$ | Root node corresponding to the full-length protein. |
| $Y$ | Observed peptide distribution from experimental data, where $Y(u)$ denotes the observed frequency or abundance of peptide $u$. |
| $P$ | Predicted (modeled) peptide distribution inferred from the degradation graph. Each component $P(u)$ is the modeled probability of absorption at node $u$. |
| $P(\cdot \mid v)$ | Marginal peptide distribution when starting from node $v$. By definition, $\mathbf{P}(\cdot \mid \Omega) = \mathbf{P}$. |
| $\mathcal{P}(v)$ | Set of parent (precursor) nodes of $v$ — i.e., peptides that can degrade into $v$. |
| $\mathcal{C}(v)$ | Set of child nodes of $v$ — i.e., peptides that can be produced from $v$. |
| $P(v \to u)$ | Cumulative probability of reaching node $u$ starting from node $v$, integrated over all possible degradation paths between $v$ and $u$. By definition, $P(v \to v) = 1$. |
| $|v|$ | Length of peptide $v$ in amino acids. |

To align the predicted distribution **P** with the observed data **Y**, we minimize the mean squared error:

$$\mathcal{L}_{\text{MSE}}(\mathbf{Y}, \mathbf{P}) = \sum_{u \in V} \left( P(u) - Y(u) \right)^2.$$

Regularization can be added to enforce sparsity or smoothness on the parameters:

$$\mathcal{L}_{\text{reg}} = \lambda_1 \sum_u \|\theta_u\|_1 + \lambda_2 \sum_u \|\theta_u\|_2^2,$$

yielding the total objective $\mathcal{L} = \mathcal{L}_{\text{MSE}} + \mathcal{L}_{\text{reg}}$.

After training, the final transition probabilities $w_{u \to v}$ are obtained via the softmax transformation of optimized $\theta$ values. Because softmax enforces strictly positive values, transition probabilities cannot be reduced exactly to zero.

### 4.3 Identifying the degradation graph using linear programming

Alternatively, inference can be formulated as a constrained flow problem where probability mass is conserved across nodes. Instead of directly learning $w_{u \to v}$, we solve for flows $F_{u \to v}$ representing the mass transferred from node $u$ to node $v$.

**Flow variables:**

$$F = \{F_{u \to v}\}, \quad \forall (u, v) \in E.$$

**Mass conservation:** At each node, the total incoming mass equals the total outgoing flow plus absorbed mass (observed abundance):

$$\sum_{v \in \mathcal{C}(u)} F_{u \to v} + Y(u) = \sum_{x \in \mathcal{P}(u)} F_{x \to u}, \quad \forall u \in V.$$

**Root injection:**

$$\sum_{v\in\mathcal{C}(\Omega)} F_{\Omega\to v} = 1.$$

**Non-negativity constraint:**

$$F_{u\to v} \geq 0, \quad \forall(u,v)\in E.$$

These constraints form a linear system that can be solved via linear programming (e.g. using PuLP [33]). From the optimized flow solution, transition probabilities are computed as normalized outgoing flows:

$$w_{u\to v} = \frac{F_{u\to v}}{\sum_{x\in\mathcal{P}(u)} F_{x\to u}}, \quad \forall(u,v)\in E.$$

The residual absorption probability at each node is defined as

$$w_{u\to u} = 1 - \sum_{v\in\mathcal{C}(u)} w_{u\to v}.$$

### 4.4 Graph neural network for graph-based inference

To demonstrate how degradation graphs can capture proteolytic patterns, we implemented a graph neural network (GNN) classifier operating on inferred degradation graphs. Each sample is represented as a directed graph $G = (V, E)$ with edge weights $w_{u\to v}$.

Each peptide node $v$ is described by a feature vector $\mathbf{x}_v \in \mathbb{R}^4$:

$$\mathbf{x}_v = \left[\text{abundance}_v \ \text{start}_v \ \text{end}_v \ \text{length}_v\right],$$

where abundance is the normalized peptide intensity, and start/end/length are position-normalized sequence descriptors.

We used a two-layer GraphConv architecture with hidden dimensionality 64 and dropout probability 0.2 after the first layer. Node embeddings were aggregated by global mean pooling to yield a graph-level representation, followed by a linear classifier with log-softmax output.

The network was trained using negative log-likelihood loss on enzyme class labels (trypsin vs. elastase), optimized with Adam (learning rate 0.005, weight decay $10^{-4}$) for 50 epochs.

### 4.5 Formal definitions of peptide distributions and degradation graphs

We now formalize peptide distributions, degradation transitions, and their graphical representation.

**Mechanistic degradation model:** Proteins degrade through sequential cleavage, producing a set of peptides. For a protein sequence $\Omega$, degradation yields peptides $\{v_1, v_2, \ldots, v_m\}$ observed with empirical frequencies $\mathbf{Y}$.

We model degradation as a probabilistic process where a peptide $v$ may remain intact or degrade into smaller peptides in $\mathcal{C}(v)$. Each transition $v \to u$ has probability $w_{v\to u}$, and self-absorption occurs with $w_{v\to v}$.

**Graph representation:** The process is represented as a directed acyclic graph $G = (V, E)$, where $V$ are peptides and $E$ are transitions ($v \to u$). Each node $v$ has incoming edges from $\mathcal{P}(v)$ and outgoing edges to $\mathcal{C}(v)$. Weights $w_{v\to u}$ satisfy $\sum_{u\in\mathcal{C}(v)\cup\{v\}} w_{v\to u} = 1$.

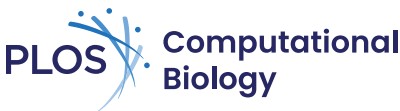

**Marginal peptide distribution:** The predicted peptide distribution **P** describes the steady-state absorption probabilities across the graph and can be computed recursively:

1. The root probability is $P(\Omega) = w_{\Omega \to \Omega}$.
2. For all other nodes:

$$P(v) = \left( \sum_{u \in \mathcal{P}(v)} P(\Omega \to u) P(u \to v) \right) w_{v \to v},$$

where $P(\Omega \to u)$ is the cumulative probability of reaching $u$ from the root. Using $P(\Omega \to u) = P(u)/w_{u \to u}$, we obtain

$$P(v) = \sum_{u \in \mathcal{P}(v)} \frac{P(u)}{w_{u \to u}} w_{u \to v} \, w_{v \to v}.$$

**Conditional distribution:** The distribution when starting from node $k$ is

$$\mathbf{P}(\cdot \mid k) = \left( \sum_{i \in \mathcal{C}(k)} \mathbf{P}(\cdot \mid i) \, w_{k \to i} \right) + \mathbf{1}_k \, w_{k \to k},$$

where $\mathbf{1}_k$ denotes a one-hot vector with unit mass at node $k$.

**Gradients:** The gradient of **P** with respect to a weight $w_{i \to j}$ is:

$$\frac{\partial \mathbf{P}}{\partial w_{i \to j}} = \begin{cases} P(\Omega \to i) \, \mathbf{P}(\cdot \mid j), & \text{if } i \neq j, \\ \mathbf{1}_i \, P(\Omega \to i), & \text{if } i = j. \end{cases}$$

**Generalization to realistic data:** In real datasets, peptide coverage is incomplete. We relax the strict two-fragment splitting model and allow each peptide to transition into any valid subsequence of itself. This yields equivalent marginal distributions **P** under simpler connectivity assumptions.

For example, if $v = ABC$, possible transitions include:

$$ABC \to \{A, B, C, AB, BC\}.$$

Weights in this simplified model can be rescaled from the idealized model to yield the same marginal distribution:

$$P(ABC \to AB) = P(ABC \to C) = \frac{w_{ABC \to \{AB, C\}}}{2}.$$

### 4.6 Formalization of key limitations and assumptions

The degradation graph formalism rests on several simplifying assumptions and biological constraints, detailed below.

**Peptide generation assumptions:** Let $Y$ represent the set of peptides observed in a peptidomic experiment. This set is the result of multiple processes acting on an initial set of peptides derived from protein degradation. We assume the following steps in the workflow:

Let $\mathbf{Y}''$ be the set of all theoretically possible peptides generated through protein degradation. This process consists of two primary operations:

- **Endoproteolytic cleavage:** The splitting of peptides within the sequence, excluding terminal amino acids. Let $E$ represent the set of such cleavages.
- **Exoproteolytic trimming:** The removal of terminal amino acids from the peptide chain. Let $X$ represent the set of terminal trimming operations.

Thus, we define $\mathbf{Y}''$ as the set resulting from the application of these two processes, where the operations $E$ and $X$ are performed on a precursor protein sequence $P$. This can be expressed as $\mathbf{Y}'' = f(P, E, X)$, where $f$ represents the function that maps the sequence $P$ under the actions of $E$ and $X$ to the set $\mathbf{Y}''$.

**Rapid degradation filter**: A biological filtering process removes some peptides in $\mathbf{Y}''$ that undergo rapid degradation for resource recycling (e.g., for energy and atoms). Let $F_1 \subseteq \mathbf{Y}''$ represent this biological filter, which results in a subset of more stable peptides denoted as $\mathbf{Y}'$. Formally, we have:

$$\mathbf{Y}' = \mathbf{Y}'' \setminus F_1,$$

where $F_1$ is the set of peptides subject to rapid degradation. The remaining set $Y'$ contains peptides that are stable.

**Peptide identification filter:** The observed set of peptides, $\mathbf{Y}$, is a subset of $\mathbf{Y}'$ because not all peptides in $\mathbf{Y}'$ are detectable through experimental techniques such as mass spectrometry. Let $F_2 \subseteq \mathbf{Y}'$ represent the filter corresponding to technical limitations (e.g., detection sensitivity, ionization efficiency, etc.), yielding the final observed peptide distribution $\mathbf{Y}$. Formally,

$$\mathbf{Y} = \mathbf{Y}' \setminus F_2,$$

where $F_2$ represents the set of peptides undetected or unquantified by the experimental apparatus.

Thus, the final observed peptide set $\mathbf{Y}$ is:

$$\mathbf{Y} = (\mathbf{Y}'' \setminus F_1) \setminus F_2$$

Additionally, peptide intensity is skewed by their ability to ionize effectively and fly in the mass spectrometer. This results in a possible inaccurate representation of $\mathbf{Y}$.

**Unidentifiability:** Approximating degradation graphs comes with fundamental limitations. For example, if $ABC$ degrades to $B$, and we observe equal abundances of $AB$ and $BC$, we cannot determine whether $B$ was degraded through $AB$ alone, $BC$ alone, or both. This issue underscores the overparametrization of degradation graphs.

### 4.7 Degradation simulation

To validate our degradation graph model and test computational methods, we developed an algorithm to simulate proteolysis. The simulator generates synthetic peptidomes by modeling endoproteolytic and exoproteolytic events according to enzyme-specific cleavage patterns.

The simulator works as follows: starting with an intact protein sequence, the simulator iteratively performs proteolytic events until a target number of peptides is generated. Each iteration randomly selects between endoproteolytic cleavage (cutting within the sequence) and exoproteolytic trimming (removing terminal residues), based on a configurable probability parameter.

For endoproteolytic events, the simulator: 1. Selects a peptide from the current pool, weighted by length and abundance 2. Determines potential cleavage sites according to the specified enzyme's regex-based rules, e.g. `(.)([K|R])([^P])(.)` for trypsin 3. Selects a primary cleavage site using the enzyme's positional preferences 4. Selects a second

cleavage site, with probability weighted by both enzyme specificity and distance from the first site (following a gamma distribution) 5. Generates up to three fragments from the two cuts

For exoproteolytic events, the simulator: 1. Randomly selects a peptide weighted by abundance 2. Removes one amino acid randomly from either the N- or C-terminus

The output is the degradation graph as well as the generated peptide distribution.

### 4.8 Application to trypsinized $\beta$-actin

We analyzed peptidomic data from trypsinized human cell lysates (PXD037803 [14]). Raw files were searched against the human proteome using PEAKS X without cleavage site restrictions, and peptide intensities were extracted and log-normalized.

For the $\beta$-actin sequence, a degradation graph was constructed and edge weights were optimized by gradient descent so that the modeled marginal distribution $\hat{Y}$ matched the observed peptide abundances $Y$. This allowed us to quantify inflows and identify bottleneck peptides, defined as intermediates with disproportionately high flow relative to their abundance.

### 4.9 Application to the diabetic uromodulin peptidome

We applied degradation graph modeling to a published urinary peptidome dataset of diabetic and healthy patients [51]. Uromodulin was selected as a case study. Peptidomic data were downloaded from ProteomeXchange (PXD012210) and searched against the human proteome without enzymatic constraints using PEAKS X. For each patient, peptide abundances corresponding to uromodulin were extracted and log-normalized. Missing peptides were imputed with the log intensity value corresponding to the lowest quantile of the data.

For each sample, a degradation graph was constructed and gradient descent was used to optimize edge weights so that the modeled marginal distribution $\hat{Y}$ matched the observed peptide abundances $Y$.

To quantify the systematic bias of traditional peptidomics, we defined the underestimation factor per sample as the total flow over the total abundance:

$$\Delta = \frac{\sum_{(u \to v) \in E} F_{u \to v}}{\sum_{v \in V \setminus \Omega} Y(v)},$$

where $F_{u \to v}$ is the total flow through edge $(u \to v)$ and $\Omega$ is the full-length protein. A value $\Delta > 1$ indicates that the total proteolytic activity is underestimated when considering peptide abundances alone.

To localize differences between groups, we further aggregated optimized inflows to peptides and mapped their $\log_2$ fold change between diabetic and healthy samples along the uromodulin backbone.

### 4.10 Application to infected hemoglobin $\alpha$ (porcine wound fluid)

We analyzed a peptidomic dataset of porcine wound fluids infected with either *Staphylococcus aureus* or *Pseudomonas aeruginosa* [11]. Peptide intensities corresponding to hemoglobin subunit $\alpha$ (HBA) were extracted and log-normalized. Missing peptides were imputed with the log intensity value corresponding to the lowest quantile of the data.

For each sample, a degradation graph was constructed and edge weights were optimized with gradient descent. As before, the total proteolytic activity per sample was quantified as the ratio between the total generated peptide flow and the sum of peptide abundances.

Group-level differences were assessed by averaging peptide inflows within infection type and calculating $\log_2$ fold changes between *S. aureus* and *P. aeruginosa* samples. These values were mapped along the HBA backbone to visualize regions of differential degradation.

## Supporting information

**S1 Fig. Relationship between degradation extent and underestimation ratio. a** Schematic illustrating how extended simulation time produces deeper and more branched degradation pathways. **b** Underestimation ratio ($\Delta$) as a function of graph size, measured by the number of edges generated during simulated proteolysis. As degradation progresses and graphs become larger, $\Delta$ increases, reflecting the growing discrepancy between total peptide intensity and the true cumulative degradation flow.
(TIFF)

## Author contributions

**Conceptualization:** Erik Hartman, Johan Malmström, Jonas Wallin.

**Data curation:** Erik Hartman.

**Formal analysis:** Erik Hartman, Jonas Wallin.

**Funding acquisition:** Johan Malmström.

**Investigation:** Erik Hartman, Jonas Wallin.

**Methodology:** Erik Hartman, Jonas Wallin.

**Project administration:** Johan Malmström, Jonas Wallin.

**Resources:** Erik Hartman, Johan Malmström.

**Software:** Erik Hartman.

**Supervision:** Johan Malmström, Jonas Wallin.

**Validation:** Erik Hartman.

**Visualization:** Erik Hartman.

**Writing – original draft:** Erik Hartman, Johan Malmström, Jonas Wallin.

**Writing – review & editing:** Erik Hartman, Johan Malmström, Jonas Wallin.

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
