## [Decision Letter · Decision Letter 0]

22 Sep 2025

PCOMPBIOL-D-25-01342

Degradation Graphs for Mechanistic Peptidomic Analysis

PLOS Computational Biology

Dear Dr. Hartman,

Thank you for submitting your manuscript to PLOS Computational Biology. After careful consideration, we feel that it has merit but does not fully meet PLOS Computational Biology's publication criteria as it currently stands. Therefore, we invite you to submit a revised version of the manuscript that addresses the points raised during the review process.

Please submit your revised manuscript within 60 days Nov 21 2025 11:59PM. If you will need more time than this to complete your revisions, please reply to this message or contact the journal office at ploscompbiol@plos.org. Please include the following items when submitting your revised manuscript:

We look forward to receiving your revised manuscript.

Kind regards,

Yanbu Guo, Ph.D.

Academic Editor

PLOS Computational Biology

Virginia Pitzer

Editor-in-Chief

PLOS Computational Biology

**Additional Editor Comments (if provided):**

Sorry for the delay. Overall, the reviewers felt the manuscript has merit, but made a number of recommendations that should be addressed in the revised version.

Furthermore, while the manuscript was submitted as a Perspective, the editors feel that it is more appropriate as a Research Article or Methods article, in line with the feedback from the reviewers. Please resubmit a revised version (taking into account the critiques) as one or other after reviewing the policies here: https://journals.plos.org/ploscompbiol/s/submission-guidelines#loc-methods-submissions

**Journal Requirements:**

**Reviewers' comments:**

Reviewer's Responses to Questions

**Comments to the Authors:**

Reviewer #1: I have the following comments.

1. It is recommended that the Introduction section should include an overview of current mainstream protein degradation models, along with an analysis of their differences and improvements compared to the proposed method. Additionally, the core innovations of this study should be clearly articulated in the Introduction.

2. The experimental validation is currently limited to simulated data and a single experimental dataset (β-actin), with a relatively small dataset size (only 70 peptides). It is suggested to broaden the validation scope by testing on diverse protein types and larger-scale real datasets.

3. The paper lacks performance comparisons between the proposed method and existing methods. It is recommended to include relevant benchmark tests to demonstrate its advantages.

4. The description of the specific implementation details for optimizing the weights of the degradation graph using gradient descent and linear programming is insufficient. It is suggested to provide algorithm pseudocode or flowcharts to enhance clarity and reproducibility.

Reviewer #2: 1 Incorporate any significant numerical findings within the abstract.

2.In the beginning, emphasise the novelties and fresh contributions of your work.

3 Give an overview of earlier research, highlighting what has been accomplished, what is still lacking, and your contributions in relation to those findings.

4 Perform further benchmark analysis to illustrate the benefits or drawbacks of your approach. To assess how much the adoption of your suggested advanced ways improves things, it would be helpful to have more comparisons with some older approaches.

5 Please give github website to release the code with datasets.

6 Please perform parameter analysis and abration experiments.

Reviewer #3: In PCOMPBIOL-D-25-01342 entitled “Degradation Graphs for Mechanistic Peptidomic Analysis”, authors proposed degradation graphs to representing proteolytic processes from peptidomic data, and show a case about how degradation graphs can be used for prediction of graph types.

It should be an interesting work to provide more computational model and biological details of protein and its dynamics in biological system. I have some questions and suggestions:

(1) The main figures should improve and provide more details about the model. For Figure 2a, as the tree structure with root of one protein, all the peptides in the tree should be given variable annotations, which can help understand the relation of variables in the model by Markov chain.

(2) Similarly, in Figure 3a, it is necessary to illustrate the common and specificity between traditional and graph methods.

(3) It is necessary to provide a pseudo-code for the computational framework in the main text.

(4) Authors have provided case application to a small experimental data, which is based on a pool data. It is necessary to provide further deep mechanistic study on larger dataset with control and case samples (e.g. normal and disease samples).

(5) It is suggested to provide some tables to summarize the current available data and method for peptidomic study.

(6) It is better to provide code and data of this study in public domain.

(7) The manuscript must be proofread to eliminate all spelling errors. Such as in Page 8, “TTo address the challenges …”

**Reviewers who wish to reveal their identities to the authors and other reviewers should include their name here (optional). These names will not be published with the manuscript, should it be accepted.**

Reviewer #1: (No Response)

Reviewer #2: None

Reviewer #3: (No Response)

**Figure resubmission:**
---

## [Decision Letter · Decision Letter 1]

1 Dec 2025

PCOMPBIOL-D-25-01342R1

Degradation Graphs Reveal Hidden Proteolytic Activity in Peptidomes

PLOS Computational Biology

Dear Dr. Hartman,

Thank you for submitting your manuscript to PLOS Computational Biology. After careful consideration, we feel that it has merit but does not fully meet PLOS Computational Biology's publication criteria as it currently stands. Therefore, we invite you to submit a revised version of the manuscript that addresses the points raised during the review process.

We look forward to receiving your revised manuscript.

Kind regards,

Yanbu Guo, Ph.D.

Academic Editor

PLOS Computational Biology

Virginia Pitzer

Editor-in-Chief

PLOS Computational Biology

**Additional Editor Comments (if provided):**

Note that two of the reviewers still have some more substantial comments that need to be address, while one has some minor comments regarding the figures. In particular, two of the reviewers ask for some comparison to alternative/simpler methods.

**Journal Requirements:**

**Reviewers' comments:**

Reviewer's Responses to Questions

**Comments to the Authors:**

Reviewer #1: I have no other comments.

Reviewer #2: The authors addressed most of my prior comments. However I have several comments as follows,

1. Explicitly distinguish degradation graphs from prior sequence-based degradation models (e.g., sequential multi-step reaction models) beyond brief mentions. Highlight unique contributions to graph structure (directed acyclic design, absorption probability) and how they solve unmet needs in dynamic proteolysis inference from static data.

2.Strengthen the rationale for choosing both gradient descent and linear programming optimization. Explain trade-offs (e.g., flexibility vs. constraint satisfaction) with empirical comparisons (e.g., convergence speed, stability across datasets) to guide users on method selection.

3. Address peptide detection bias: Discuss how incomplete peptide coverage (a key limitation) impacts graph inference and propose mitigation strategies (e.g., imputation methods tailored to degradation logic, or sensitivity analyses on missing peptides).

4. Expand the machine learning section: Test the GraphConv model on more proteases (beyond trypsin/elastase) and real-world datasets (not just simulated data) to validate its ability to capture protease-specific signatures in complex samples.

5. Improve figure annotations: For key figures (e.g., Fig. 2, 4), add labels for biological relevance (e.g., "bottleneck peptide linked to inflammation") and clarify how graph features map to experimental observations.

Reviewer #3: Authors have responded to all my concerned questions, and made certain revisions.

There are a few remaining points:

(1) Fig3f, the numbers in x-axis should be labeled clearly.

(2) Fig5c, there should be P value of significance test of group differences between Diabetes and Healthy.

(3) Fig6c, there should be P value of significance test of group differences between S.aureus and P.aeruginosa.

Reviewer #4: This study achieves significant breakthroughs in the analysis of protein dynamic degradation by constructing an innovative computational framework called the 'degradation graph.' For the first time, this method models protein degradation as a directed acyclic graph (DAG) and introduces the probabilistic flow analysis paradigm (Figures 2a–c), successfully addressing the systematic underestimation problem in traditional proteomics analyses caused by neglecting the order of degradation. By proposing the dual concepts of 'absorption probability' and 'flow quantification,' combined with parameter sensitivity analysis and a multi-dimensional validation system (simulated data in Figure 4, in vitro β-actin data in Figure 3, and diabetic urine and infected wound fluid data in Figures 5–6), the method significantly improves the quantification accuracy of protease activity. Notably, during the revision process, the authors added two critical clinical datasets on diabetes and wound infection and supplemented technical details such as algorithm pseudocode, effectively addressing the initial review comments.

Based on the current quality of the manuscript, it is recommended to address the following issues before acceptance:

1. Although the authors state on page 8 that “there is currently no directly comparable computational framework,” the manuscript lacks a systematic comparison with any simplified baseline methods, making it difficult to intuitively demonstrate the quantitative advantage of the degradation graph. Please include in the supplementary materials a comparison experiment with the conventional "peptide intensity summation method," i.e., directly using observed peptide abundances as protease activity estimates and comparing with the total flow estimates from the degradation graph, further quantifying the "3–4 fold underestimation" mentioned in Figures 3f, 5c, and 6c.

2. Page 18 mentions the problem of "Unidentifiability," but the stability of the model under different initializations or noisy conditions is not systematically evaluated. Please show this as much as possible in the methods or figures, such as the coefficient of variation of edge weights from multiple runs of gradient descent on the same dataset.

3. Please adjust the clarity of textual content in the figures and use vector graphics wherever possible.

This study is highly innovative and practical in terms of conceptual, methodological, and application aspects, has adequately addressed the reviewer comments from the initial draft, and has made significant progress in both theoretical and experimental validation.

I recommend revisions before acceptance, mainly addressing the three points mentioned above to provide further clarification or analysis, thereby enhancing the rigor and persuasiveness of the manuscript.

**Have the authors made all data and (if applicable) computational code underlying the findings in their manuscript fully available?**

Reviewer #1: Yes

Reviewer #2: Yes

Reviewer #3: Yes

Reviewer #4: None

PLOS authors have the option to publish the peer review history of their article (what does this mean?). If published, this will include your full peer review and any attached files.

Reviewer #1: No

Reviewer #2: No

Reviewer #3: No

Reviewer #4: No

**Figure resubmission:**
---

## [Decision Letter · Decision Letter 2]

2 Feb 2026

Dear Mr Hartman,

We are pleased to inform you that your manuscript 'Degradation Graphs Reveal Hidden Proteolytic Activity in Peptidomes' has been provisionally accepted for publication in PLOS Computational Biology.

Best regards,

Yanbu Guo, Ph.D.

Academic Editor

PLOS Computational Biology

Virginia Pitzer

Editor-in-Chief

PLOS Computational Biology

Reviewer's Responses to Questions

**Comments to the Authors:**

Reviewer #1: I have no other comments.

Reviewer #2: none

Reviewer #3: Authors have responded to all my concerned questions, and made a revision well.

Reviewer #4: I have carefully reviewed the revised manuscript and the point-by-point response. Overall, the authors have adequately addressed my previous concerns, and I therefore recommend acceptance. Specifically, the authors provided simulation-based quantification showing that the underestimation ratio (Δ) increases with degradation extent/peptidome complexity, thereby more clearly establishing the quantitative advantage over the conventional peptide-intensity summation approach. The revision also includes additional evaluations of unidentifiability and model stability, including analyses of the coefficient of variation of edge weights across different random initializations, as well as sensitivity sweeps over learning rates and training epochs demonstrating that the main conclusions and top features remain consistent once convergence is achieved. These additions substantially strengthen the comparability, robustness, and overall persuasiveness of the method.

**Have the authors made all data and (if applicable) computational code underlying the findings in their manuscript fully available?**

Reviewer #1: None

Reviewer #2: Yes

Reviewer #3: Yes

Reviewer #4: None

PLOS authors have the option to publish the peer review history of their article (what does this mean?). If published, this will include your full peer review and any attached files.

Reviewer #1: No

Reviewer #2: No

Reviewer #3: No

Reviewer #4: No

---

## [Editor Report · Acceptance letter]

PCOMPBIOL-D-25-01342R2

Degradation Graphs Reveal Hidden Proteolytic Activity in Peptidomes

Dear Dr Hartman,

I am pleased to inform you that your manuscript has been formally accepted for publication in PLOS Computational Biology. Your manuscript is now with our production department and you will be notified of the publication date in due course.

With kind regards,

Livia Horvath
